# Systemic Effects of Homoarginine Supplementation on Arginine Metabolizing Enzymes in Rats with Heart Failure with Preserved Ejection Fraction

**DOI:** 10.3390/ijms241914782

**Published:** 2023-09-30

**Authors:** Petra Büttner, Sarah Werner, Julia Böttner, Susann Ossmann, Edzard Schwedhelm, Holger Thiele

**Affiliations:** 1Department of Cardiology, Heart Center Leipzig at University of Leipzig, 04289 Leipzig, Germany; 2Department of Cardiac Surgery, Heart Center Leipzig at University of Leipzig, 04289 Leipzig, Germany; 3Institute of Clinical Pharmacology and Toxicology, University Medical Center Hamburg-Eppendorf, 20251 Hamburg, Germany; 4DZHK (German Centre for Cardiovascular Research), Partner Site Hamburg/Kiel/Lübeck, 20246 Hamburg, Germany

**Keywords:** heart failure with preserved ejection fraction, homoarginine, arginase, NO synthase, dimethylarginine dimethylaminohydrolase, glycine amidinotransferase

## Abstract

A restoration of low homoarginine (hArg) levels in obese ZSF1 rats (O-ZSF1) before (S1-ZSF1) and after (S2-ZSF1) the manifestation of heart failure with preserved ejection fraction (HFpEF) did not affect the worsening of cardiac HFpEF characteristics. Here, potential regulation of key enzymes of arginine metabolism in other organs was analyzed. Arginase 2 (ARG2) was reduced >35% in the kidney and small intestine of hArg-supplemented rats compared to O-ZSF1. Glycine amidinotransferase (GATM) was 29% upregulated in the kidneys of S1-ZSF1. Dimethylarginine dimethylaminohydrolase 1 (DDAH1) levels were reduced >50% in the livers of O-ZSF1 but restored in S2-ZSF1 compared to healthy rats (L-ZSF1). In the skeletal muscle, iNOS was lower in O-ZSF1 and further decreased in S1-ZSF1 and S2-ZSF1 compared to L-ZSF1. iNOS levels were lower in the liver of the S2-ZSF1 group but higher in the kidneys of S1-ZSF1 compared to L-ZSF1. Supplementation with hArg in an in vivo HFpEF model resulted in the inhibition of renal ARG2 and an increase in GATM expression. This supplementation might contribute to the stabilization of intestinal iNOS and ARG2 imbalances, thereby enhancing barrier function. Additionally, it may offer protective effects in skeletal muscle by downregulating iNOS. In the conceptualization of hArg supplementation studies, the current disease progression stage as well as organ-specific enzyme regulation should be considered.

## 1. Introduction

Regulation of vasodilation strongly depends on the activity of nitric oxide synthases (NOS) and the accessibility of the substrate arginine, which is also metabolized by competing arginases (ARG). While NOS has a higher affinity for arginine, the activity of ARG is higher and thus both enzymes are thought to have comparable substrate utilization rates [1]. Next to arginine, homoarginine (hArg) is also a weak NOS substrate that inhibits ARG [2]. Importantly, low hArg levels independently predict cardiovascular mortality whereby the molecular mechanisms underpinning this observation are unknown [3]. Endothelial vasodilatory dysfunction is considered as one causal factor in heart failure with preserved ejection fraction (HFpEF), characterized by preserved systolic cardiac function but diastolic dysfunction, increased vascular and left ventricular stiffness and impaired relaxation. HFpEF patients suffer from dyspnea, exercise intolerance, and are subjected to high morbidity and mortality [4,5]. Noteworthily, higher concentrations of hArg were associated with milder HFpEF symptoms and improved exercise tolerance [6]. Thus, hArg may represent a therapeutic agent that was not tested in clinical setting yet. Human data are sparse but 24 healthy volunteers who took 125 mg hArg daily had no adverse effects and also no changes in vascular and neuronal function [2]. This issue was recently addressed in an HFpEF rat model. ZSF1 rats are F1 hybrid cross breeds from male spontaneously hypertensive rats and female Zucker diabetes rat strains with two different leptin receptor mutations. Compound heterozygous offspring of these rats eat excessively, become obese, and spontaneously develop hypertension, hyperlipidemia, glucose intolerance, exercise intolerance, and finally HFpEF [7,8]. These rats, like human HFpEF patients, have significantly lower hArg levels compared to healthy controls [9]. Consequently, they were supplemented with hArg to reestablish the physiological hArg levels. However, no effects on cardiac characteristics of HFpEF manifestation were observed, such as increased left ventricular volume, mass, and stiffness, nor the associated decrease in specific skeletal muscle force [10]. Noteworthily, HFpEF is regarded as a multi-organ disease [11,12] and thus, although hArg supplementation has no effect on cardiac characteristics, there may be effects on other organs. This assumption is underpinned by the observation that exercise can improve the cardiorespiratory performance of HFpEF patients while cardiac function is not improved [13]. In addition to the previously mentioned NOS and arginases, arginine and its derivatives, such as the NOS-inhibiting methylarginines, are metabolized by dimethylarginine dimethylaminohydrolase (DDAH1), alanine glyoxylate transaminase-2 (AGXT2), and glycine amidinotransferase (GATM) [3]. Finally, key enzymes in arginine-metabolism are particularly more important in other organs than the heart, especially the liver and kidney. Thus, gene and protein expression of inducible (iNOS) and endothelial NOS (eNOS), ARG1 and ARG2, and DDAH1, AGXT2, and GATM were determined in the liver, kidney, small intestine, aorta, and skeletal muscle of ZSF1 rats with HFpEF that received hArg supplementation.

## 2. Results

As previously reported, obese (O-ZSF1) rats develop symptoms of HFpEF and experience a significant hArg deficiency ([9,10], Figure 1, Table 1). The hArg supplementation normalized the serum concentrations in S1/S2-ZSF1 compared to O-ZSF1 at an age of 32 weeks (Figure 1, Table 1). In contrast, the serum concentration of asymmetric dimethylarginine (ADMA) was found 18% elevated in O-ZSF1 rats (*p* = 0.025) and 16% in S2-ZSF1 (*p* = 0.041) as compared with lean control rats (L-ZSF1). Comparable concentrations of arginine but also ornithine and citrulline, the turnover products of ARG, DDAH and NOS, were observed in all four experimental groups (Table 1).

Gene expression of GATM was increased in the kidneys of O-ZSF1 (*p* = 0.042) and S1-ZSF1 (*p* = 0.05) compared to L-ZSF1, and this was confirmed at the protein level in S1-ZSF1 (*p* = 0.083 vs. L-ZSF1, *p* = 0.098 vs. O-ZSF1). DDAH1 protein levels in the liver were decreased by 72% in O-ZSF1 (*p* = 0.003) and by 54% in S1-ZSF1 (*p* = 0.026) compared to L-ZSF1. Levels in S2-ZSF1 were normalized to L-ZSF1 levels. In the small intestine, DDAH1 protein content was reduced by 37% in S1-ZSF1 compared to L-ZSF1 (*p* = 0.008) and by 29% compared to O-ZSF1 (*p* = 0.064), while O-ZSF1 and S2-ZSF1 levels were comparable to L-ZSF1. Gene expression of DDAH1 was also reduced in the small intestine of S1-ZSF1 compared to O-ZSF1 (*p* = 0.048) and S2-ZSF1 (*p* = 0.073). In the kidney, a numerical reduction of 13% of DDAH1 compared to L-ZSF1 was observed (*p* = 0.089). ARG2 protein levels in the kidney were reduced by >35% in S1-ZSF1 and S2-ZSF1 (both *p* = 0.001) compared to O-ZSF1. In the small intestine, ARG2 was reduced by 40% in S1-ZSF1 (*p* = 0.064) and by 53% in S2-ZSF1 (*p* = 0.012) compared to O-ZSF1 (Figure 1, Table 1). Gene expression of ARG2 was comparable in these organs and in skeletal muscle in all experimental groups.

In the liver, the small intestine and skeletal muscle iNOS protein content was decreased by 18% (*p* = 0.070), 23% (*p* = 0.054), and 31% (*p* = 0.003), and in aorta, it was 85% higher (*p* = 0.077) in O-ZSF1 compared to L-ZSF. In the liver, iNOS was normalized in the S1-ZSF1 group (*p* = 0.568), while a reduction of 24% was still seen in the S2-ZSF1 group (*p* = 0.013 vs. L-ZSF1). In the skeletal muscle, hArg supplementation caused an iNOS protein reduction of 47% in S1-ZSF1 and a reduction of 59% in S2-ZSF1 (both *p* < 0.0001 vs. L-ZSF1). In the small intestine, hArg supplementation normalized the iNOS level. At a gene expression level, we observed a strong downregulation of iNOS in the kidney (Table 1), while on a protein level, we observed numerically increased expression by 50% in O-ZSF1 (*p* = 0.061) and S2-ZSF1 (*p* = 0.060) and 72% in S1-ZSF1 (*p* = 0.005).

Gene and protein expression of ARG1 in the liver, AGXT2 in kidney and liver, and eNOS in aorta were unchanged (Figure 1, Table 1).

## 3. Discussion

A worse outcome in cardiovascular disease is independently associated with low hArg levels indicating a causative connection between this arginine analogue, arginine-metabolizing enzymes, and potentially endothelial dysfunction, but the underpinning mechanisms are elusive [14]. Supplementation with hArg of obese ZSF1 with HFpEF restored physiological blood levels without affecting arginine concentration, which is comparable to findings from healthy humans [2]. Nevertheless, hArg supplementation could not prevent manifestation and progression of HFpEF in terms of cardiac characteristics in this rat model [10]. However, this may not be surprising as the expression of arginine metabolizing enzymes is comparably low in the heart itself. Alterations in arginine metabolites more likely have secondary effects that favor HFpEF development; for example, limited arginine accessibility for eNOS in cardiac arteries [3]. Furthermore, the kidney and liver in particular, as well as the small intestine, skeletal muscle, and blood vessels, express different arginine-consuming enzymes and thus may be affected by altered hArg levels. Indeed, concerning the regulation of arginine-metabolizing enzymes in O-ZSF1 rats with low hArg and in O-ZSF1 rats supplemented with hArg, we found that (1) GATM was upregulated in obese rats supplemented with hArg before detectable symptoms of HFpEF, and (2) that DDAH1 was lower in the livers of O-ZSF1 rats but was normalized by hArg supplementation initiated upon the manifestation of HFpEF. In the small intestine and numerically in the kidney, hArg supplementation starting before HFpEF manifests resulted in downregulation of DDAH1. Furthermore, it was also found that (3) ARG2 was downregulated in kidneys and small intestines of O-ZSF1 with hArg supplementation, and (4) that iNOS was sensitive to low hArg levels in O-ZSF1 rats but was also regulated when hArg was supplemented with different patterns in the liver, kidney, skeletal muscle small intestine, and aorta.

Low homoarginine has been identified as an independent predictor of morbidity and mortality in patients suffering from heart failure [15,16,17]. GATM in rats is predominantly expressed in the kidney [9] and utilizes arginine to produce hArg, thus controlling endogenous hArg levels [2]. GATM-knock out mice show a homoarginine and creatine deficiency leading to left ventricular dysfunction [18] and this phenotype can be rescued by homoarginine supplementation [19]. Of particular note, heart failure patients with decreased circulating hArg also had an increased GATM expression in the left ventricle that was normalized after mechanical unloading and recovery of the left ventricle [20]. Rats receiving hArg from an early age (S1-ZSF1) had an increased GATM gene and protein expression, which is in contrast to recent findings in mice where GATM was downregulated when hArg was supplemented, implying end-product inhibition [21]. Importantly, GATM synthesizes creatine, which is significantly involved in energy supply of cardiac and skeletal muscle [21]. However, that hArg supplementation might increase creatine level needs to be determined.

ADMA, released during degradation of proteins with methylated arginine residues, is a competitive NOS inhibitor and higher concentrations predict adverse cardiovascular outcomes [3]. DDAH1 metabolizes and thus defuses ADMA [3]. ADMA concentrations were higher in O-ZSF1 compared to L-ZSF1 while, simultaneously, DDAH1 expression was significantly lower in the liver. Effects of hArg supplementation on DDAH1 were organ specific. In the small intestine of S1-ZSF1 rats that received hArg at an early age before HFpEF manifested, DDAH1 expression was lower than in O-ZSF1 rats. On the other hand, DDAH1 levels in the liver and small intestine were normalized only in obese rats in which hArg supplementation started after the manifestation of HFpEF (S2-ZSF1).

The majority of total body ARG activity and thus arginine consumption takes place in the form of ARG1 activity in the urea cycle in the liver, but this enzyme was affected neither by a lack in hArg nor by its supplementation. In the contrary, the second isoform ARG2, which controls arginine availability, especially for the production of cardiovascular relevant vasoactive NO [1], was downregulated in the kidney and the small intestine in animals supplemented with hArg. An inhibition of ARG1 by hArg was reported previously [2]. Noteworthily, in the kidney, ARG2 upregulation has been associated with ischemia–reperfusion-induced acute kidney injury and the identification of a specific inhibitor was suggested [22]. According to our data, hArg may be suitable. An imbalance in vasoactive NO causing peripheral endothelial dysfunction is supposed to underpin HFpEF [23]. Thus, eNOS expression was analyzed in aorta but no differences between the experimental groups were observed. However, it was observed that iNOS exhibited a significant reliance on the hArg level. While eNOS and the neural NO synthase are continuously expressed, iNOS expression is induced in response to stimulation. An upregulation of iNOS with subsequent massively increased NO synthesis was described in inflammation, obesity, and type 2 diabetes, conditions that are found in O-ZSF1, and the upregulation is thought to induce muscular insulin resistance and mitochondrial dysfunction [24,25]. Surprisingly, in the liver, small intestine, and skeletal muscle, iNOS was numerically downregulated in O-ZSF1 with low hArg. Intestinal homeostasis, mucosal integrity as well as intestinal barrier function are vulnerable to an imbalance of iNOS and ARG activity, whereas lowering of ARG, as seen in the small intestines of S1- and S2-ZSF1, was found to prolong murine life span [26]. Supposedly, hArg supplementation may thus be beneficial for small intestine homeostasis in HFpEF. This is of special interest as HFpEF patients have a higher circulating TMAO level, a metabolite that predicts adverse outcome in HFpEF [27], which originates from the gut and indicates altered intestine mucosa integrity [28]. HFpEF-associated detrimental alterations in skeletal muscle function and structure are not prevented by hArg supplementation [10]. Low hArg levels in HFpEF were associated with a decreased iNOS level in the skeletal muscle and this decrease was further amplified by hArg supplementation. In the skeletal muscle, NO produced by iNOS can quickly reach toxic levels [29] and thus the downregulation of iNOS, for example, observed after long-term training, is discussed as a protective mechanism [24]. Accordingly, iNOS downregulation in skeletal muscle may represent a beneficial side effect of hArg supplementation, especially if it is combined with exercise, which was already found to improve the cardiorespiratory performance in HFpEF without improving cardiac characteristics [13]. Interestingly, the human iNOS gene was initially described in hepatocytes that responded to inflammatory stimuli whereas iNOS-derived NO protects the liver during sepsis and ischemia–reperfusion by blocking apoptosis and decreasing hepatotoxicity [30]. Supplementation with hArg starting at an early age (S1-ZSF1) reduced the iNOS downregulation observed in O-ZSF1 indicating hepato-protective abilities of hArg. In the kidney, iNOS protein content was higher in rats under hArg supplementation, whereas on a transcriptional level a downregulation was observed. This contradictory observation cannot be clarified at present, but poor correlation between transcriptional and translational levels is frequently observed [31] and iNOS in particular is regulated transcriptionally and post-transcriptionally through a complex network of transcription factors, non-coding RNAs, and post-translational modification [32].

We recently reported alterations in enzymes from arginine metabolism in O-ZSF1 rats at an age of 20 weeks. The protein level of ARG1 in the liver and ARG2 in the kidney were 2.5-fold and 1.6-fold downregulated at an age of 20 weeks in O-ZSF1, respectively [9]. Here, rats were 32 weeks at sacrifice and supposedly in end-stage HFpEF [33] and ARG1 and ARG2 protein levels were comparable to L-ZSF1. Furthermore, the effects on the analyzed enzymes differed depending on whether hArg supplementation was started before or after the manifestation of cardiac HFpEF features. Likewise, only S2-ZSF1 rats had normalized DDAH1 level in the liver, while S1-ZSF1 rats were comparable with O-ZSF1. These contradictive observations might be explained by organ-specific involvement at various progression stages in the multi-organ disease HFpEF [12]. In summary, age as well as the HFpEF progression stage shall be noted as important cofactors in HFpEF-associated alterations in arginine metabolism in different organs and for conceptualization of hArg supplementation studies.

## 4. Materials and Methods

Eighteen female obese F1 hybrid rats (O-ZSF1) (ZSF1-Leprfa Leprcp/Crl, Charles River, IN, USA) and five lean ZSF1 rats (L-ZSF1) serving as controls were studied. At an age of six weeks, the O-ZSF1 animals were randomly grouped into (1) no hArg supplementation (O-ZSF1, *n* = 6), (2) immediate hArg supplementation (drinking water enriched with hArg 70 mg/L) (S1-ZSF1, *n* = 6), and (3) hArg supplementation starting at an age of 13 weeks when HFpEF was manifest according to echocardiography (S2-ZSF1, *n* = 6) [10]. Water and chow were provided ad libitum. At an age of 32 weeks, animals were sacrificed in deep anesthesia, blood was collected from the beating heart; organs were dissected following a standardized sample collection routine and weighted. Defined organ parts were snap-frozen in liquid N_2_, stored at −80 °C, pulverized, and used for RNA and protein analysis. A total of 30 mg were used for RNA isolation using the RNeasy Kit and QIAShredder (Qiagen, Hilden, Germany) according to the manufacturer’s recommendation. RNA concentration was photometrically determined and 250 or 500 ng specimens were reverse transcribed using Omniscript RT kit with poly dT Primer (Qiagen, Hilden, Germany). Quantitative Realtime-PCR used Takyon Mastermix (Eurogentec, Seraing, Belgium) and self-designed primers against Agxt2 (fw: ccgccagcctcttcctaaaa, rv:cttccagctttgtgacacgc), Arg1 (fw:cccgcagcattaaggaaagc, rv:tgaaaggggctgtcattggg), Arg2 (fw:gggcagcctctttcctttct, rv:gcaggctccacatctcgtaa), Ddah1 (fw:tgacaagctcactgtaccgg, rv:acctttgcgctttctgggta), Gatm (fw:cgcagagaagccaggttaca, rv:ggagagcacaagaccaggtc), iNos (fw:cggtgcggtcttttcctatg, rv:cagagtcttgtgcctttggg) and eNos (fw:agaactcttcactctgcccc, rv:tattggacacagctgggagg). Hypoxanthine phosphoribosyltransferase 1 was determined as “housekeeper” (fw:cccagcgtcgtgattagtga, rv:ggcctcccatctccttcatg). For protein analysis, 10–20 mg of frozen sample was homogenized in RIPA buffer containing a protease and a phosphatase inhibitor mix (Serva, Heidelberg, Germany) and sonicated. Protein concentration was determined using the BCA method (bicinchoninic acid assay, Pierce, Bonn, Germany). Antibodies were purchased from Cell Signaling (Danvers, MA, USA) Arginase 1 (93668) and Arginase 2 (55003), Thermo Fisher (Waltham, MA, USA) GATM (PA5-76957), DDAH1 (PA5-50610), iNOS (PA1-036), and AGXT2 (PA5-103587), and Abcam (Cambridge, UK) eNOS (ab76198). As “housekeeper proteins”, alpha-Tubulin (ab7291) from Abcam and Glyceraldehyde-3-phosphate dehydrogenase (5G4) from HyTest (Turku, Finland) were determined.

All experiments are in line with ARRIVE guidelines and were approved by the local Animal Research Council, University of Leipzig and the Landesbehörde Sachsen (TVV 40/19). Established and validated protocols for LC-MS/MS were used to assess serum Arg, ornithine, citrulline, ADMA, and hArg concentrations as recently reported [9]. Data were normally distributed and ANOVA with Tukey post hoc testing was used considering a *p*-value ≤ 0.05 as statistically significant while obvious numerical differences with *p*-value ≤ 0.1 were also included in the discussion with regards to the small sample size and the resulting implications for Type I and Type II errors [34].

## 5. Conclusions

Pathologically low hArg levels of HFpEF, as well as restoration of physiological levels by hArg supplementation in an HFpEF rat model, result in extensive regulation of key enzymes of arginine metabolism, particularly iNOS, ARG2, GATM, and DDAH1, with organ-specific patterns. The effects on organ function and the role of the dysregulation in the development of HFpEF are currently unclear. This systemic impact of hArg needs to be considered in all disease conditions underpinned by low hArg levels and where hArg supplementation is discussed as potential interventional strategy.

## 6. Limitations

Data presented in this study reflect alterations in progressed HFpEF only. Other time points should be analyzed in a comparable way. The pancreas, although it is an important GATM-expressing organ, was not accessible for this study. Enzyme activity was not analyzed. Pronounced inter-individual variation leading to presumed, since numerically traceable, but not significant differences was observed. This should be addressed by increased sample numbers.

## Figures and Tables

**Figure 1 ijms-24-14782-f001:**
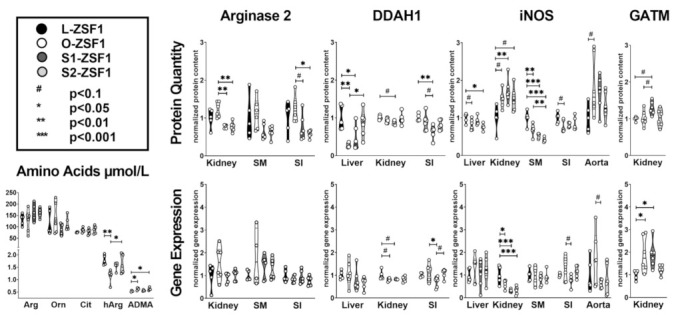
Protein quantity determined by Western blot analysis and gene expression measured by quantitative Realtime-PCR of Arginase 2, DDAH1 (dimethylarginine dimethylaminohydrolase 1), GATM (glycine amidinotransferase), and iNOS (inducible NO synthase) in obese ZSF1 rats with HFpEF (O-ZSF1), and O-ZSF1 supplemented with hArg starting before (S1-ZSF1) and after (S2-ZSF1) detectable cardiac HFpEF characteristics. Protein and gene expression were normalized to values determined in healthy rats (L-ZSF1). Serum level of arginine (Arg), ornithine (Orn), citrulline (Cit), homoarginine (hArg), and asymmetric dimethylarginine (ADMA) were determined by mass spectroscopy. SI—small intestine, SM—skeletal muscle.

**Table 1 ijms-24-14782-t001:** Summary of body and organ weight at sacrifice, serum concentrations of amino acids, results from Western blot analysis, and gene expression measurements of key enzymes in arginine metabolism in obese ZSF1 rats with HFpEF (O-ZSF1) and O-ZSF1 supplemented with hArg starting before (S1-ZSF1) and after (S2-ZSF1) detectable cardiac HFpEF characteristics. Protein and gene expression were normalized to values determined in healthy rats (L-ZSF1). Abbreviations: ADMA (asymmetric dimethyl arginine), ASXL1 (alanine glyoxylate transaminase-2), DDAH1 (dimethylarginine dimethylaminohydrolase 1), eNOS (endothelial NO synthase), GATM (glycine amidinotransferase), iNOS (inducible NO synthase), SI—small intestine, SM—skeletal muscle.

		L-ZSF1	O-ZSF1	S1-ZSF1	S2-ZSF1	L-ZSF1 vs. O-ZSF1	L-ZSF1 vs. S1-ZSF1	L-ZSF1 vs. S2-ZSF1	O-ZSF1 vs. S1-ZSF1	O-ZSF1 vs. S2-ZSF1
Weight	Body weight (g)	283 ± 9	529 ± 23	533 ± 24	513 ± 21	<0.001	<0.001	<0.001	0.760	0.236
	Heart (g)	1.16 ± 0.12	1.56 ± 0.15	1.71 ± 0.10	1.58 ± 0.09	0.002	<0.001	<0.001	0.098	0.854
	Liver (g)	9.49 ± 0.30	23.71 ± 3.76	26.05 ± 1.28	23.39 ± 1.92	<0.001	<0.001	<0.001	0.197	0.859
	Kidney (g)	1.10 ± 0.04	1.69 ± 0.16	1.78 ± 0.21	1.68 ± 0.11	<0.001	<0.001	<0.001	0.429	0.902
Serum	Arginine (umol/L)	132 ± 25	127 ± 48	164 ± 37	161 ± 21	0.995	0.507	0.586	0.347	0.418
	Citrulline (umol/L)	76 ± 2	86 ± 11	79 ± 13	87 ± 17	0.655	0.984	0.570	0.825	0.999
	Homoarginine (umol/L)	1.79 ± 0.2	1.15 ± 0.27	1.50 ± 0.22	1.64 ± 0.39	0.008	0.527	0.812	0.113	0.045
	Ornithine (umol/L)	119 ± 55	139 ± 65	89 ± 24	114 ± 31	0.910	0.753	0.998	0.352	0.838
	ADMA (umol/L)	0.53 ± 0.03	0.63 ± 0.05	0.56 ± 0.03	0.62 ± 0.06	0.025	0.818	0.041	0.131	0.992
Western Blot results	Arginase 1 liver	1.00 ± 0.19	0.81 ± 0.52	1.01 ± 0.44	1.27 ± 0.23	0.842	1.000	0.653	0.795	0.191
	Arginase 2 kidney	1.00 ± 0.25	1.22 ± 0.20	0.78 ± 0.05	0.77 ± 0.13	0.172	0.167	0.141	0.001	0.001
	Arginase 2 SM	1.00 ± 0.56	1.11 ± 0.40	0.63 ± 0.20	0.64 ± 0.16	0.955	0.345	0.357	0.126	0.131
	Arginase 2 SI	1.00 ± 0.42	1.24 ± 0.35	0.74 ± 0.35	0.58 ± 0.10	0.610	0.559	0.172	0.064	0.012
	AGXT2 liver	1.00 ± 0.39	0.90 ± 0.25	0.93 ± 0.16	0.82 ± 0.17	0.907	0.966	0.615	0.996	0.935
	AGXT2 kidney	1.00 ± 0.14	1.02 ± 0.25	1.12 ± 0.10	1.05 ± 0.18	0.999	0.684	0.966	0.747	0.988
	DDAH1 liver	1.00 ± 0.33	0.28 ± 0.05	0.46 ± 0.32	0.85 ± 0.35	0.003	0.026	0.809	0.697	0.014
	DDAH1 kidney	1.00 ± 0.05	0.93 ± 0.06	0.87 ± 0.05	0.97 ± 0.15	0.532	0.089	0.942	0.633	0.832
	DDAH1 SI	1.00 ± 0.09	0.89 ± 0.22	0.63 ± 0.17	0.80 ± 0.13	0.680	0.008	0.230	0.064	0.806
	eNOS aorta	1.00 ± 0.16	1.43 ± 0.37	1.53 ± 0.45	1.43 ± 0.49	0.303	0.164	0.302	0.978	1.000
	GATM kidney	1.00 ± 0.06	1.02 ± 0.21	1.29 ± 0.18	1.02 ± 0.23	0.997	0.083	0.997	0.098	1.000
	iNOS liver	1.00 ± 0.13	0.82 ± 0.14	0.91 ± 0.11	0.76 ± 0.08	0.070	0.568	0.013	0.518	0.148
	iNOS kidney	1.00 ± 0.34	1.50 ± 0.23	1.72 ± 0.32	1.50 ± 0.32	0.061	0.005	0.060	0.600	0.606
	iNOS SM	1.00 ± 0.19	0.69 ± 0.15	0.53 ± 0.06	0.41 ± 0.06	0.003	<0.001	<0.001	0.155	0.006
	iNOS SI	1.00 ± 0.14	0.77 ± 0.15	0.81 ± 0.06	0.83 ± 0.17	0.054	0.144	0.209	0.948	0.868
	iNOS aorta	1.00 ± 0.37	1.85 ± 0.81	1.67 ± 0.46	1.32 ± 0.38	0.077	0.211	0.770	0.935	0.348
Gene Expression	Arginase 1 liver	1.00 ± 0.50	1.78 ± 1.34	0.77 ± 0.44	0.88 ± 0.46	0.388	0.961	0.994	0.155	0.233
	Arginase 2 kidney	1.00 ± 0.52	1.55 ± 0.74	0.92 ± 0.21	1.03 ± 0.24	0.254	0.993	1.000	0.136	0.260
	Arginase 2 SM	1.00 ± 0.21	1.80 ± 1.27	1.41 ± 0.51	1.30 ± 0.45	0.300	0.737	0.878	0.792	0.642
	Arginase 2 SI	1.00 ± 0.28	0.83 ± 0.14	0.94 ± 0.28	0.77 ± 0.20	0.624	0.973	0.353	0.841	0.957
	AGXT2 liver	1.00 ± 0.44	1.71 ± 0.79	1.22 ± 0.55	1.26 ± 0.58	0.251	0.932	0.892	0.518	0.588
	AGXT2 kidney	1.00 ± 0.55	2.15 ± 0.98	1.95 ± 1.38	1.58 ± 0.67	0.235	0.389	0.761	0.984	0.734
	DDAH1 liver	1.00 ± 0.19	1.09 ± 0.54	0.78 ± 0.48	0.66 ± 0.26	0.982	0.799	0.523	0.547	0.285
	DDAH1 kidney	1.00 ± 0.20	0.81 ± 0.07	0.82 ± 0.02	0.83 ± 0.12	0.058	0.092	0.101	0.995	0.990
	DDAH1 SI	1.00 ± 0.09	1.12 ± 0.38	0.71 ± 0.23	1.09 ± 0.20	0.855	0.259	0.933	0.048	0.996
	eNOS aorta	1.00 ± 0.81	0.86 ± 0.65	1.26 ± 1.04	1.68 ± 1.60	0.997	0.978	0.743	0.921	0.587
	GATM kidney	1.00 ± 0.21	1.92 ± 0.78	1.89 ± 0.59	1.22 ± 0.21	0.042	0.050	0.900	1.000	0.131
	iNOS liver	1.00 ± 0.29	1.32 ± 0.59	1.06 ± 0.60	1.25 ± 0.56	0.761	0.998	0.873	0.833	0.995
	iNOS kidney	1.00 ± 0.38	0.61 ± 0.18	0.31 ± 0.06	0.32 ± 0.17	0.042	<0.001	<0.001	0.116	0.138
	iNOS SM	1.00 ± 0.27	0.96 ± 0.44	0.79 ± 0.24	0.92 ± 0.23	0.997	0.649	0.966	0.811	0.996
	iNOS SI	1.00 ± 0.08	1.27 ± 0.53	0.77 ± 0.23	1.15 ± 0.25	0.532	0.641	0.867	0.065	0.921
	iNOS aorta	1.00 ± 0.73	1.73 ± 1.16	0.62 ± 0.19	0.80 ± 0.67	0.417	0.844	0.974	0.090	0.190

## Data Availability

Data are available on request.

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
