# Peer review of "Systemic Effects of Homoarginine Supplementation on Arginine Metabolizing Enzymes in Rats with Heart Failure with Preserved Ejection Fraction"

_ijms, 2023, doi:10.3390/ijms241914782_

Round 1

Reviewer 1 Report (Previous Reviewer 2)

The revision is satisfactory.

Author Response

Thank you very much for your efforts.

Reviewer 2 Report (New Reviewer)

Major comments:

1. In the article, the authors aimed to explore the systemic effects of hArg supplementation in rats with HFpEF, especially in the kidney, liver, small intestine, as well as skeletal muscle system. But, as Table 1 shown, the weight of the body, heart, liver, and kidney has no difference in the rats with HFpEF after hArg supplementation. Furthermore, the gene- and protein expression of key enzymes of arginine almost has no change between O-ZSF1, S1-ZSF1, and S2-ZSF1. So, the present data have failed to support the viewpoint that the systemic effect of hArg supplementation on arginine metabolizing enzymes in rats with HFpEF.

2. It would be better if the arginine metabolism mechanism and key enzymes can be appropriately presented by a flow chart to facilitate a better comprehension for readers in the manuscript.

3. It is curious as to why the change in iNOS gene expression levels and protein levels in the kidney tend to be opposite as shown in Table 1, please elaborate on it further.

4. In lines 193-195, the authors described that “low hArg level in HFpEF associate with decreased iNOS level in skeletal muscle and this decrease cannot be prevented but is further amplified by hArg supplementation”, and skeletal iNOS downregulation might represent a beneficial side effect of hArg supplementation. From this statement, both of hArg deficiency and supplementation are associated with decreased iNOS, which tends to confuse the reader.

5. In the article, the authors highlighted that hArg supplementation does not influence cardiac characteristics. It is suggested that the author supplement relevant experience to further support the standpoint.

Minor comments:

 1. It would be helpful if the authors display Figure 1 as a histogram to make the difference between the four experimental groups clearer.

2. The quality of the Western Blot image about iNOS in aorta samples needs to be improved.

Minor editing of the English language is required.

Author Response

Dear Editorial Board, Dear Reviewer,

Thank you very much for your ongoing efforts to assess and improve our manuscript and your comments.

Please find our responses below.

Major comments:

  1. In the article, the authors aimed to explore the systemic effects of hArg supplementation in rats with HFpEF, especially in the kidney, liver, small intestine, as well as skeletal muscle system. But, as Table 1 shown, the weight of the body, heart, liver, and kidney has no difference in the rats with HFpEF after hArg supplementation. Furthermore, the gene- and protein expression of key enzymes of arginine almost has no change between O-ZSF1, S1-ZSF1, and S2-ZSF1. So, the present data have failed to support the viewpoint that the systemic effect of hArg supplementation on arginine metabolizing enzymes in rats with HFpEF.

-> The current study originally was not conducted to demonstrate systemic effects – see line 57: “may be effects on other organs”. The primary endpoints were cardiac characteristics, but hArg had no effect (line 54). As written in the introduction (line 64) arginine is metabolized in different organs. The Reviewer correctly points out that organ weights did not differ between the three experimental obese groups. However, this does not necessarily mean that enzyme expression is comparable. Indeed, protein concentration of arginase 2 in kidney and small intestine was comparable between lean control rats and obese rats not receiving hArg but was down regulated in both obese groups receiving hArg (see figure 1). Interestingly, iNOS down regulation in skeletal muscle of obese rats was further down regulated by hArg supplementation. In conclusion, hArg regulates different enzymes in different organs, justifying the phrase “systemic effect”.

  1. It would be better if the arginine metabolism mechanism and key enzymes can be appropriately presented by a flow chart to facilitate a better comprehension for readers in the manuscript.

-> We discussed this suggestion but decided not to include a further figure. The presentation of the study results was planned in a dense format as communication and thus we mentioned excellent references to provide the interested reader with further information. The very complex interactions in arginine metabolism are comprehensively described by Morris (doi:10.3945/jn.115.226621). To address the comment we added a sentence in line 61 where we referenced the review of Morris.

  1. It is curious as to why the change in iNOS gene expression levels and protein levels in the kidney tend to be opposite as shown in Table 1, please elaborate on it further.

-> Thank you for this interesting comment. Indeed, we cannot explain the observation at the current stage. We added a paragraph in line 208 to discuss this observation.

Noteworthy, it was discussed before that “the correlation between expression levels of protein and mRNA in mammals is relatively low, with a Pearson correlation coefficient of ~0.40” (https://bmcgenomics.biomedcentral.com/articles/10.1186/s12864-017-3683-9 and https://pubmed.ncbi.nlm.nih.gov/22411467/).

  1. In lines 193-195, the authors described that “low hArg level in HFpEF associate with decreased iNOS level in skeletal muscle and this decrease cannot be prevented but is further amplified by hArg supplementation”, and skeletal iNOS downregulation might represent a beneficial side effect of hArg supplementation. From this statement, both of hArg deficiency and supplementation are associated with decreased iNOS, which tends to confuse the reader.

-> Thank you for this comment. We rewrote the paragraph to improve the readability: “Low hArg level in HFpEF were associated with decreased iNOS level in skeletal muscle and this decrease was further amplified by hArg supplementation. In skeletal muscle NO produced by iNOS can quickly reach toxic levels and thus the down-regulation of iNOS as, for example, observed after long-term training, is discussed as a protective mechanism [19]. Accordingly, iNOS down regulation in skeletal muscle may represent a beneficial side effect of hArg supplementation, especially if it is combined with exercise, which was already found to improve the cardiorespiratory performance in HFpEF without improving cardiac characteristics [10].”

  1. In the article, the authors highlighted that hArg supplementation does not influence cardiac characteristics. It is suggested that the author supplement relevant experience to further support the standpoint.

-> The study presented here does not focus on the heart. The heart was mentioned because the homoarginine supplementation trial was initially conducted to study a potential effect of hArg supplementation on cardiac function in rats with HFpEF. The results were published in an open access journal as cited in the introduction: “no effect on cardiac characteristics of HFpEF manifestation and the associated decrease in specific skeletal muscle force were observed [8].” In our opinion, a discussion of cardiac functional characteristics is beyond the scope of the current study. To address the reviewer's suggestion, we have added the phrase "namely increased left ventricular volume, mass and stiffness," in line 55.

Minor comments:

  1. It would be helpful if the authors display Figure 1 as a histogram to make the difference between the four experimental groups clearer.

 -> Initially, we decided to present individual data to allow the interpretation of inter-individual variance and to provide maximal transparency. To address the Reviewer comment we now changed the graphs to truncated violin plots with individual points. We think that this data presentation provides both a general overview and inter-individual variance.

  1. The quality of the Western Blot image about iNOS in aorta samples needs to be improved.

-> We repeated the Western Blot analysis and observed comparable data as included in the manuscript (please see pdf). Please note that only four lean animals were analyzed, as aortic sample was no longer available for one individual.

This manuscript is a resubmission of an earlier submission. The following is a list of the peer review reports and author responses from that submission.

Round 1

Reviewer 1 Report

The article is very interesting. It raises the role of homoarginine in the activation of enzymes with low levels in organs related to cardiac problems. However, there are some points that should be clarified: 

In materials and methods, describe the methodology of each assay as it appears in the results table.

I would appreciate to detail the methodology of the Western Blot and present the photographs of the development (Bands) with the respective controls.

Please adapt the conclusion, it is very ambiguous about what was found in the present work.

Reference 1 should be adequate according to the format of the journal. Review the rest of the references.

Author Response

The article is very interesting. It raises the role of homoarginine in the activation of enzymes with low levels in organs related to cardiac problems. However, there are some points that should be clarified: 

In materials and methods, describe the methodology of each assay as it appears in the results table.

  • Thank you for your evaluation. We added description in the header of figure 1 (line 91) and also in Material and methods (line 236-252)

I would appreciate to detail the methodology of the Western Blot and present the photographs of the development (Bands) with the respective controls.

  • We prepared a pdf including the Western blot images for your evaluation.

 Please adapt the conclusion, it is very ambiguous about what was found in the present work.

  • We rewrote the conclusion (line 262-268)

Reference 1 should be adequate according to the format of the journal. Review the rest of the references.

  • Thank you for the comment. References were checked and corrected.

Reviewer 2 Report

The study titled " systemic effects of homoarginine supplementation on arginine metabolizing enzymes in rats with heart failure with preserved ejection fraction" by Buttner et al is interesting. The authors have studied the effects of homoarginine supplementation in obese rats at early stage before onset of HFpEF and after the manifestation of HFpEF. They did not observe beneficial effects of the supplementation in the heart however the analysis of the key enzymes in arginine metabolism in other organs were analyzed and presented and discussed. The results showed are convincing. However, it critical to improve the manuscript on the following points.

1) In Figure 1, could you mention the technique used of protein analysis? The methods section says vaguely about mass spectrometry, but it is not clearly stated whether the protein analysis was by mass spectrometry analysis or by western blotting. If by western blotting please include the representative blots in the manuscript. In Table 1, it says western blot results and therefore it is required to include the representative blots as a figure with the quantification data for the readers to appreciate the results completely. Also include more details on the western blotting including antibody details in the methods section. Also, include the primer details of the RNA study and the method of quantification. 

2) The results presented in the Figure1 and Table 1 is very exhausting for the reader to grasp easily what to look for. It would be great if you can summarize the results in the form of a graphical abstract using some pictorial representations of each organ and the enzymes and indicating the upregulation or downregulation under each condition will be very helpful for the reader to understand the overall findings at a glance. It would be little more appealing. This a suggestion to improve the readership as the data are too descriptive and difficult to arrive at a meaningful conclusion. 

Minor changes:

1) Table 1, change the spelling of "Weigth" to "Weight". 

2) In Discussion, Line 120-122: "Supplementation....humans" needs to be rephrased for correctness.

3) Line 159-161: " On the...level" needs to be rephrased for correctness.

Small typographical errors are noted and commented.  

Author Response

Reviewer 2

The study titled " systemic effects of homoarginine supplementation on arginine metabolizing enzymes in rats with heart failure with preserved ejection fraction" by Buttner et al is interesting. The authors have studied the effects of homoarginine supplementation in obese rats at early stage before onset of HFpEF and after the manifestation of HFpEF. They did not observe beneficial effects of the supplementation in the heart however the analysis of the key enzymes in arginine metabolism in other organs were analyzed and presented and discussed. The results showed are convincing. However, it critical to improve the manuscript on the following points.

-> Thank you very much for your evaluation and the helpful comments.

1) In Figure 1, could you mention the technique used of protein analysis? The methods section says vaguely about mass spectrometry, but it is not clearly stated whether the protein analysis was by mass spectrometry analysis or by western blotting.

  • We added description in the header of figure 1 (line 91) and also in Material and methods (line 236-252)

If by western blotting please include the representative blots in the manuscript.

  • We prepared a pdf including the Western blot images for your evaluation.

In Table 1, it says western blot results and therefore it is required to include the representative blots as a figure with the quantification data for the readers to appreciate the results completely. Also include more details on the western blotting including antibody details in the methods section. Also, include the primer details of the RNA study and the method of quantification. 

  • -> Thank you for your comments. We applied changes to the manuscript, please see Material and methods (line 236-252)

2) The results presented in the Figure1 and Table 1 is very exhausting for the reader to grasp easily what to look for. It would be great if you can summarize the results in the form of a graphical abstract using some pictorial representations of each organ and the enzymes and indicating the upregulation or downregulation under each condition will be very helpful for the reader to understand the overall findings at a glance. It would be little more appealing. This a suggestion to improve the readership as the data are too descriptive and difficult to arrive at a meaningful conclusion. 

-> We agree that the results reported in this study are comprehensive. Nevertheless, we focused on key enzymes and analyzed them in all organs that are involved in arginine metabolism. To decrease the information’s in figure 1 we already excluded the results for AGXT2, eNOS and Arg1 as these enzymes were not regulated by hArg supplementation. We think that figure 1 is of importance to those readers who want to study the inter-individual variance. To address your comment we prepared a graphical abstract.

Minor changes:

1) Table 1, change the spelling of "Weigth" to "Weight". 

-> Thank you for this comment – fixed.

2) In Discussion, Line 120-122: "Supplementation....humans" needs to be rephrased for correctness.

-> Thank you for this comment – rephrased.

3) Line 159-161: " On the...level" needs to be rephrased for correctness.

-> Thank you for this comment – rephrased.

Round 2

Reviewer 2 Report

The authors do not show the loading control for most of the protein in various tissue except for the first one in the supplementary figure in the revised manuscript. This is not acceptable, and it is not convinced about the way the protein expression levels were quantified without the loading control from the same blots. Although the authors included the suggestions and corrections, the lack of appropriate loading control blots for various tissues makes the study results invalid at this point. 

The author's do not show Figure 2 in the manuscript even though it is written in the discussion that it is shown (line 142).

Not applicable.